# Strategic HIV Case Findings among Infants at Different Entry Points of Health Facilities in Cameroon: Optimizing the Elimination of Mother-To-Child Transmission in Low- and- Middle-Income Countries

**DOI:** 10.3390/v16050752

**Published:** 2024-05-10

**Authors:** Celine Nguefeu Nkenfou, Georges Nguefack-Tsague, Aubin Joseph Nanfack, Sylvie Agnes Moudourou, Marie-Nicole Ngoufack, Leaticia-Grace Yatchou, Elise Lobe Elong, Joel-Josephine Kameni, Aline Tiga, Rachel Kamgaing, Nelly Kamgaing, Joseph Fokam, Alexis Ndjolo

**Affiliations:** 1Chantal BIYA International Reference Centre for research on HIV/AIDS Prevention and Management, Yaoundé P.O. Box 3077, Cameroon; a_nanfack@yahoo.fr (A.J.N.); sylvie.moudourou@gmail.com (S.A.M.); laeticiagrace@gmail.com (L.-G.Y.); elongelise@yahoo.fr (E.L.E.); kamenikjoel@gmail.com (J.-J.K.); tigaaline@yahoo.fr (A.T.); r.kamgaing@yahoo.it (R.K.); kmnelly2007@gmail.com (N.K.); andjolo@yahoo.com (A.N.); 2Higher Teacher Training College, University of Yaoundé I, Yaoundé P.O. Box 3077, Cameroon; 3Faculty of Medicine and Biomedical Sciences, University of Yaoundé I, Yaoundé P.O. Box 3077, Cameroon; nguefacktsague@yahoo.fr; 4Faculty of Sciences, University of Yaoundé I, Yaoundé P.O. Box 3077, Cameroon; mnngoufack@yahoo.fr; 5Faculty of Health Sciences, University of Buea, Buea P.O. Box 63, Cameroon

**Keywords:** pediatric HIV, entry points, testing coverage, positivity rate, multiple Poisson regression analysis

## Abstract

Background: HIV case finding is an essential component for ending AIDS, but there is limited evidence on the effectiveness of such a strategy in the pediatric population. We sought to determine HIV positivity rates among children according to entry points in Cameroon. Methods: A facility-based survey was conducted from January 2015 to December 2019 among mother–child couples at various entry points of health facilities in six regions of Cameroon. A questionnaire was administered to parents/guardians. Children were tested by polymerase chain reaction (PCR). Positivity rates were compared between entry points. Associations were quantified using the unadjusted positivity ratio (PR) for univariate analyses and the adjusted positivity ratio (aPR) for multiple Poisson regression analyses with 95% confidence intervals (CIs). *p*-values < 0.05 were considered significant. Results: Overall, 24,097 children were enrolled. Among them, 75.91% were tested through the HIV prevention of mother-to-child transmission (PMTCT) program, followed by outpatient (13.27%) and immunization (6.27%) services. In total, PMTCT, immunization, and outpatient services accounted for 95.39% of children. The overall positivity was 5.71%, with significant differences (*p* < 0.001) between entry points. Univariate analysis showed that inpatient service (PR = 1.45; 95% CI: [1.08, 1.94]; *p* = 0.014), infant welfare (PR = 0.43; 95% CI: [0.28, 0.66]; *p* < 0.001), immunization (PR = 0.56; 95% CI: [0.45, 0.70]; *p* < 0.001), and PMTCT (PR = 0.41; 95% CI: [0.37, 0.46]; *p* < 0.001) were associated with HIV transmission. After adjusting for other covariates, only PMTCT was associated with transmission (aPR = 0.66; 95% CI: [0.51, 0.86]; *p* = 0.002). Conclusions: While PMTCT accounts for most tested children, high HIV positivity rates were found among children presenting at inpatient, nutrition, and outpatient services and HIV care units. Thus, systematic HIV testing should be proposed for all sick children presenting at the hospital who have escaped the PMTCT cascade.

## 1. Introduction

The early infant diagnostic (EID) program for HIV aims to screen all HIV-exposed children during the first postnatal visit (at 4–6 weeks) for timely treatment initiation for every infected child [1,2]. Nevertheless, less than 20% of HIV-exposed infants in resource-limited settings (RLSs) are enrolled in the EID program, indicating poor progress toward eliminating the mother-to-child transmission of HIV (eMTCT) [1,2,3,4,5]. Testing exposed infants comes at the end of the PMTCT cascade. This cascade comprises a series of crucial stepwise activities that form a critical pathway to the successful prevention of mother-to-child transmission (PMTCT). The cascade begins with all pregnant women and ends with the determination of the final HIV status of HIV-exposed infants (HEIs), which is set at 18 months of age. The simplified cascade includes the following steps: determining the number of pregnant women expected at antenatal clinics, identifying those who were actually received and tested for HIV, determining the number of those who tested positive, identifying those who were initiated on treatment, determining the expected number of exposed children born to HIV-positive mothers, identifying those who were born or registered at a health facility and received prophylactic nevirapine, identifying those who were tested by PCR at 6 weeks, determining the number of those who tested positive, and identifying those children who were initiated on treatment. It is worth noting that the gaps at each stage are narrowing year by year. However, the gap in coverage for the early diagnosis of EID remains significant. Previous studies have shown high rates of loss to follow-up (LTFU) throughout PMTCT cascade care and subsequent risks of vertical transmission at the community level [6,7,8,9,10,11]. LTFU is defined as the mother–infant missing appointed visits for more than three months [9]. These LTFU couples may have transferred their care services elsewhere, died, or dropped out for other reasons. High rates of loss to follow-up in the prevention of mother-to-child transmission (PMTCT) of HIV programs in Cameroon will only contribute to hindering the success of the program and are associated with unfavorable outcomes among exposed babies. Once children have been lost from the PMTCT/EID system, there are very few opportunities for testing or re-entry into care. Strategies have been developed to reduce loss to follow-up. These strategies include using community-based organizations to trace or locate mothers or children who have dropped out, using a trained community health relay agent for home visits, and providing integrated services directly in the community with healthcare staff. Of note, in Cameroon, there are two EID testing systems, which include a standard reference laboratory and point-of-care (POC) EID testing systems, that cover all 4623 PMTCT sites at the country level. At the start of this program, only centralized reference laboratories equipped with high-volume testing platforms (96-well plates) were used. The major drawback of this platform was the long turnaround time between blood collection and the return of results to the patient. This drawback was associated with an increase in exposed children lost to follow-up. In 2017, point-of-care (POC) tests were introduced to overcome this shortcoming. These POC tests can process 1 to 4 samples at a time, and results are available in approximately 1 h and 30 min. This allows for the results to be given to the patient on the same day, providing a way to reduce losses to follow-up.

According to global reports, only 59% of infants exposed to HIV received an HIV test within the first 2 months of their lives by the end of 2018 [12]. This observation, therefore, underscores the poor coverage of EID globally, especially in resource-limited settings (RLSs), where most HIV-exposed/infected children and adolescents are tested only when clinically ill, thus limiting the chances of treatment success beyond three months of life. According to Cameroonian national reports, HIV EID coverage is 65% [13] (CNLS, 2016), and it has been reported as low elsewhere [11,14,15,16]. In the same national report, antiretroviral treatment (ART) coverage in the pediatric population was the lowest in the country compared with other target population groups (adults: 65%, pregnant women: 75%, children and adolescents: 45%). These observations stress the need to establish strategies to enhance HIV testing coverage in children to achieve the expected goals of eliminating pediatric HIV by 2030.

Several comprehensive and differentiated strategies for identifying undiagnosed children have been developed and implemented [17,18,19]. Of note, most national guidelines have limited pediatric components in this regard, focusing mostly on voluntary counseling and testing (VCT), provider-initiated testing and counseling (PITC), and opt-out testing for adults [20,21]. Few studies have specifically looked at multifaceted strategies for improving the case findings of HIV-exposed and HIV-infected children, and in Cameroon, only one previous study was conducted in a limited number of health facilities [22]. This finding outlines a simple, two-tiered approach to identifying children at specific healthcare entry points. The study in [22], evaluating the positivity rate according to service points from POC EID systems in Cameroon, showed that more than half of infected children were identified from non-PMTCT service points [22]. Regarding the limited coverage from the aforementioned studies, as well as the changing dynamics in PMTCT/EID programs over time, it is, therefore, of paramount importance to ensure the effectiveness of the testing strategy is monitored, to assess the contribution of various entry points on the testing coverage of EID, and to delineate the positivity load as per the potential entry points within a large set of health facilities within the national health system of Cameroon. With the goal of contributing to an informed strategy for HIV testing uptake in the national EID/PMTCT program of Cameroon, we, herein, evaluated the HIV testing yield among exposed/infected children and their positivity rates according to entry points based on a conventional EID testing system.

## 2. Materials and Methods

### 2.1. Study Design and Settings

In this facility-based survey across six regions of Cameroon covered by the Chantal BIYA International Reference Centre for Research on HIV/AIDS Prevention and Management in Cameroon (CIRCB), early infant diagnosis (EID) data were collected over five years (from January 2015 to December 2019) and were analyzed according to the entry points of health facilities. The EID program started in Cameroon in 2009 with CIRCB as the main reference laboratory.

### 2.2. Description of the Reference Study Site

The CIRCB is a government institution of the Ministry of Public Health dedicated to HIV research and patient monitoring in several aspects, including (a) HIV early infant diagnosis in the frame of the national PMTCT program; (b) diagnosis of coinfections with HIV; (c) viral load measurement; (d) CD4 and CD8 T lymphocyte counts; (e) biochemical and hematological tests for drug safety; and (f) genotypic HIVDR testing (GRT) at subsidized costs, with quality control programs conducted in partnership with Quality Assessment and Standardization of Indicators (QASI) and other international agencies. Within the frame of the national EID program, the CIRCB covers 6 out of 10 regions in Cameroon, namely, the Adamawa, Centre, East, North, Far North, and South Regions of the country (http://www.circb.cm/btc_circb/web/) accessed 20 February 2024.

### 2.3. Data Collection

As of 2015, the nationwide questionnaire was revised and updated to include several questions, one of which regarded entry points at health facilities. The following entry points were identified: besides PMTCT, the major entry points are outpatient (consultation), inpatient (hospitalization), nutrition, vaccination (immunization), adult HIV care units, and the infant welfare units. Children and infants included in this study were from 6 out of 10 regions in Cameroon: Adamawa, Centre, East, North, Far North, and South. Children/infants aged less than 18 months are eligible for the HIV early infant diagnosis program based on polymerase chain reaction (PCR). HIV exposure was confirmed directly from the mother, if present, or by serologic antibody testing of the child. This serologic test was performed according to the national algorithm. Briefly, the blood sample is tested with Determine HIV ½ (Alere Medical Co., Ltd., Chiba, Japan); if reactive, the sample is retested with a more specific HIV test, in this case, OraQuick (OraSure Technologies, Inc., Bethlehem, PA, USA). If reactive, the test is considered positive for the presence of HIV antibodies. In this case, if the child is less than 18 months old, the blood sample is collected DBSs and tested by PCR. Sociodemographic and clinical data were collected for each infant and from their mothers. Blood was collected from each infant or child as dried blood spots and sent to the reference laboratory, where HIV was tested using virological methods. Briefly, EID was performed at the reference laboratory by using the Abbott m2000RT platform. The Abbott RealTime HIV-1 qualitative platform is an in vitro amplification assay for the qualitative detection of human immunodeficiency virus type 1 (HIV-1) nucleic acids from human plasma and dried blood spots (DBSs) (Abbott Molecular Inc., Libertyville, IL, USA). In this study, Abbott’s qualitative real-time PCR was used to detect HIV-1 nucleic acids. The lower detection limit of the assay is 2500 copies/mL. If the copy number is ≥2500 copies/mL, the test detects the presence of nucleic acids, and the result is “detected”. If the copy number is less than 2500 copies/mL [23,24], the test does not detect HIV nucleic acids, and the result is “not detected”. This high detection limit and low sensitivity is a drawback of this test, especially in the Option B+ era of PMTCT. Option B+ offers lifelong ART to pregnant women, regardless of CD4 count, during pregnancy, childbirth, and breastfeeding. To overcome this limitation, children are followed up at 18 months of age and tested serologically before their HIV status is declared.

Data were entered into Microsoft Excel 2016, cleaned, and then exported for analyses using STATA version 14. Frequencies and percentages (%) were used to describe qualitative variables. Pearson’s chi-square test (Fisher’s exact test where relevant) was used to establish relationships between qualitative variables. Since this was a cross-sectional study, associations were further quantified using the unadjusted positivity ratio (PR) for univariable (bivariate) analysis and the adjusted positivity ratio (aPR) for multiple Poisson regression analysis with a 95% confidence interval (CI), as recommended by Barros and Hirakata [25]. Variables used for adjustment included the sex of the child, child’s age, mother’s age, region, service, year of testing, feeding option (from the collected data, we identified three feeding options: exclusive breastfeeding, exclusive artificial feeding, and mixed feeding), whether the PMTCT protocol was respected (if the mother received anti-retroviral during pregnancy), and whether the child received cotrimoxazole prophylaxis. A *p*-value of less than 0.05 was considered statistically significant.

### 2.4. Ethical Considerations

The present study was conducted in accordance with the Declaration of Helsinki adopted by the 18th World Medical Assembly in 1964 with respect to international regulations for ethics and good clinical practices. As a nested study of the “MOBABY” project on mother-to-child HIV transmission in Cameroon, ethical clearance was obtained from the National Ethics Committee for Research on Human Health (reference N° 2013/11/375/L/CNERSH/SP). Administrative authorizations were obtained from the Directorate General of the CIRCB and the directors of participating health facilities. Mothers provided their informed consent. Data from the CIRCB EID database and health facilities were deidentified and anonymized for the purposes of confidentiality. EID results were provided to mothers at no cost. HIV-positive infants were promptly started on antiretroviral therapy for their clinical benefits. Mothers of HIV-infected children were advised to follow their own treatment plans and ensure that their babies also received treatment as recommended by medical professionals. Mothers of HIV-negative infants were advised to follow PMTCT cascade care to prevent HIV transmission, particularly during breastfeeding. They were also advised to adhere to their own treatment plans and seek medical advice if either they or their children feel sick. We note that HIV treatment has been free of charge in Cameroon since 2007.

## 3. Results

### 3.1. Characteristics of the Study Population

A total of 24,121 participants were enrolled from 2015 to 2019. Among them, 12,209 (50.92%) were girls, and 11,912 (49.38%) were boys. Half (50.83%) of the children were aged between 6 weeks and 6 months at the time of testing. Other age groups were less represented. Mothers aged 25–29 years were the most represented (29.40%), while only 3.51% of mothers were between 40 and 45 years old. Most of the participants (32.76%) were enrolled in 2016 compared with other years. Nearly half of the participants (47.01%) were recruited in the central region during the enrollment period (Table 1).

### 3.2. Entry Points

Seven entry points were recorded in the study: outpatient (CS), inpatient (HO), infant welfare care (IC), nutrition (NU), immunization/vaccination (PE), PMTCT (PT), and HIV care units (UPs). The number of children tested differed according to these entry points (Table 2). Of the 24,121 enrolled children, 24,097 had information about their entry points. Among them, 18,292 (75.91%) were tested through PT, which is the main entry point, followed by outpatient (13.27%) and immunization services (6.27%). Thus, PMTCT, immunization services, and outpatient services accounted for 95.39% of children. The other entry points contributed less (0.68–1.84%).

### 3.3. HIV Status According to the Entry Point

Globally, the entry point was associated with HIV transmission (*p* < 0.001; Table 3). The lowest likelihood of transmission occurred at the PMTCT entry point (4.58%). Children enrolled in this service had a lower likelihood of being infected than those enrolled in outpatient services (PR = 0.41; 95% CI: [0.37–0.46]). The likelihood of transmission was highest in children recruited from inpatient services. Among those children recruited at inpatient services, a total of 219 (83.91%) were HIV-uninfected, while 42 (16.09%) were HIV-infected. These children were 1.45 times more likely to be infected compared with those coming from outpatient services (PR = 1.45; 95% CI: [1.08–1.94]).

While PMTCT, immunization services, and outpatient services accounted for 95.39% of children, HIV positivity was higher for inpatient services (16.09%), nutrition (11.59%), outpatient services (11.12%), and HIV care units (11.06%). These 4 later accounted for nearly 16% of children.

Univariate analysis showed that inpatient services (PR = 1.45; 95% CI: [1.08, 1.94]; *p* = 0.014), infant welfare (PR = 0.43; 95% CI: [0.28, 0.66]; *p* < 0.001), immunization (PR = 0.56; 95% CI: [0.45, 0.70]; *p* < 0.001), and PMTCT (PR = 0.41; 95% CI: [0.37, 0.46]; *p* < 0.001) were associated with HIV transmission. After adjusting for other covariates, only PMTCT was associated with transmission (aPR = 0.66; 95% CI: [0.51, 0.86]; *p* = 0.002) (see Table 4).

## 4. Discussion

Our study found that, while PMTCT accounts for most tested children, high HIV positivity rates can be found among children presenting at inpatient services, nutrition services, outpatient services, and HIV care units. A strategy of collecting and using data for Cameroon’s national improvement program was used. The data collected and presented in this study utilize provider-initiated testing and counseling at various entry points. This is one of the five key strategies for strengthening HIV case finding and linkage to treatment for infants, children, and adolescents [19].

The overall positivity rate of this study was 5.71%, far from achieving MTCT elimination as defined by the WHO. PMTCT impact is measured by MTCT at 6 weeks and 18 months postpartum. MTCT rates <5% in breastfeeding populations and <2% in non-breastfeeding populations should be achieved. In 2014, the WHO, in collaboration with the Joint United Nations Program on HIV/AIDS (UNAIDS), the United Nations Population Fund (UNFPA), and the United Nations Children’s Fund (UNICEF), published two impact criteria and three process criteria to validate eMTCT [26,27]. One of the problems identified in Cameroon as preventing eMTCT is the low coverage of EID and the poor identification of infected children.

Our findings showed that PMTCT is still the main entry point for EID [19,22], where 75.93% of children are tested. The WHO recommends testing infants for HIV with unknown status admitted to inpatient, nutrition, outpatient, and immunization services [28], which was adopted and recommended by Cameroon national guidelines [29]. It has also been shown that universal maternal HIV screening at immunization visits with referral to EID and maternal ART initiation may reduce MTCT [30]. The WHO recommendations have been published, but they have not yet been widely implemented or put into practice. A high and wide range of children lost to follow-up from the PMTCT cascade [7,31,32,33,34,35] may only show up for care as a result of sickness. In the fight against HIV, the Joint United Nations Program on HIV/AIDS (UNAIDS) has established the 95-95-95 targets, whereby 95% of PL-HIV should be diagnosed, 95% of those diagnosed with HIV should receive antiretroviral therapy (ART), and 95% of all those receiving ART should achieve viral suppression. To reach the “first 95” in pediatrics, we ought to expand the coverage of testing through the extension of entry points and active searches for exposed children [36]. Improving the testing of women before they become pregnant is an important corollary to the EID program, in addition to universal maternal HIV screening. This can be achieved through restructured reproductive health services, which will include systematic HIV testing for adolescent girls and young women presenting for reproductive health services. This is part of the first of four pillars of PMTCT service delivery representing the cornerstones of comprehensive PMTCT: the primary prevention of HIV infection among women of childbearing age.

Altogether, inpatients and outpatients allowed us to identify 23.47% (373/1589) of infected children. Without much effort, proposing EID testing for infants and children presenting at outpatient services will allow for the identification of a great portion of exposed and infected children after PMTCT.

Other studies have shown that inpatient and nutrition services are where infected children are found [37]. Expanding routine early infant diagnosis screening beyond the traditional PMTCT setting to outpatient and inpatient entry points will increase the identification of HIV-infected infants. Simply testing all children presenting in the hospital, despite the mother’s HIV status, can yield great results.

Previous studies have shown a high prevalence of HIV (8.8%) among children aged 5–9 years in rural areas of Cameroon [38]; meanwhile, those cases identified with PCR accounted for approximately 5%, meaning that a high number of children escape the PMTCT cascade, and based on a National Aids Control Committee (NACC) report, only 69% of children have access to EID testing [39], which is even lower worldwide [40]. It will thus be important to find them elsewhere, with priority given to services where they present as sick.

Predicted problems in systematic testing implementation at identified testing points may be related to cost and the availability of human resources due to increased workload, but proper and strategic planning could overcome these limitations.

## 5. Limitations of this Study

This study did not include children referred from the community using advanced identification strategies (index case finding), as this entry point was not listed in the PCR request form. Additionally, this study did not consider the testing conducted at 18 months of age using a rapid test, which is recommended in order to complete the PMTCT cascade for an exposed child before exiting the program.

## 6. Conclusions

Careful reflection on appropriate testing strategies, such as systematic testing at the consultation and hospitalization ward, should be considered as a strategy for the identification of exposed infected children. These findings may help HIV care programs significantly identify infected children and improve access to early pediatric treatment. This strategy will enable the identification of those children who escape the PMTCT cascade, as well as those whose mothers did not present at a health facility during pregnancy. While identifying infected children, this strategy will also identify their infected mothers. The abovementioned strategy should be applied to all infants and children admitted to inpatient or outpatient services.

## Figures and Tables

**Table 1 viruses-16-00752-t001:** Distribution of child and mother characteristics according to sex, age, region, and year.

Characteristics	Number	Percent (%)
**Child’ sex**		
Girl	12,209	50.62
Boy	11,912	49.38
**Child’ age**		
<6 weeks	5759	24.03
[6 weeks–6 months]	12,180	50.83
[6 months, 18 months]	5702	23.79
[18 months–30 months]	323	1.35
**Mother’s age**		
15–19	1278	5.69
20–24	4339	19.29
25–29	6614	29.40
30–34	6334	28.15
35–39	3141	13.96
40–45	790	3.51
**Region**		
Adamawa	2317	9.49
Centre	11,476	47.01
East	3325	13.62
Far North	1814	7.43
North	3154	12.92
South	2325	9.52
**Year**		
2015	3656	14.98
2016	7997	32.76
2017	6735	27.59
2018	3753	15.37
2019	2270	9.30

**Table 2 viruses-16-00752-t002:** Distribution of children according to entry points.

Entry Point	Number	Percentage (%)
Outpatient services	3198	13.27
Inpatient services	262	1.09
Infant welfare	443	1.84
Nutrition	164	0.68
Immunization services	1512	6.27
PMTCT	18,292	75.91
HIV care unit	226	0.94

**Table 3 viruses-16-00752-t003:** HIV status according to entry point.

Entry Point (*p* = 0.000)	Negative N (%) 22,967 (94.15)	Positive N (%) 1426 (5.85)	Total 24,393	PR (95%)	*p*
Outpatient services	2838 (88.88)	355 (11.12)	3193	1	
Inpatient services	219 (83.91)	42 (16.09)	261	1.45 (1.08–1.94)	0.014
Infant welfare	421 (95.25)	21 (4.75)	442	0.43 (0.28–0.66)	0.000
Nutrition	145 (88.41)	19 (11.59)	164	1.04 (0.67–1.61)	0.85
Immunization services	1416 (93.71)	95 (6.29)	1511	0.56 (0.45–0.70)	0.000
PMTCT	17,444 (95.42)	838 (4.58)	18,282	0.41 (0.37–0.46)	0.000
HIV care unit	201 (88.94)	25 (11.06)	226	0.99 (0.68–1.46)	0.98

**Table 4 viruses-16-00752-t004:** HIV status according to the entry point after adjusting for other covariates (sex of the child, child’s age, mother’s age, region, service, year, deeding, PMTCT protocol, and under cotrimoxazole).

	aPR (95% CI)	*p*
Service		
Outpatient services	1	
Inpatient services	1.44 (0.86, 2.40)	0.16
Infant welfare	0.77 (0.32, 1.81)	0.545
Nutrition	0.42 (0.16, 1.09)	0.074
Immunization services	0.77 (0.51, 1.18)	0.235
PMTCT	0.66 (0.51, 0.86)	0.002
HIV care unit	0.55 (0.24, 1.28)	0.166

## Data Availability

All data generated or analyzed during this study are included in this published article.

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
