# Peer review of "Strategic HIV Case Findings among Infants at Different Entry Points of Health Facilities in Cameroon: Optimizing the Elimination of Mother-To-Child Transmission in Low- and- Middle-Income Countries"

_viruses, 2024, doi:10.3390/v16050752_

Round 1

Reviewer 1 Report

Comments and Suggestions for Authors

This is an important study of case finding optimization for early infant HIV diagnosis in Cameroon. I have a few suggestions to improve communication with global audiences.

Page 2: “…only one study was conducted in fewer health facilities”.  Consider, “…only one previous study was conducted in a limited number of health facilities” if 58 health facilities is “limited”. Add reference 22 to this sentence. The next sentence doesn’t need “Tchendjou et al, 2020” since it refers to ref 22.

Page 3: Is the “HIV care unit” for adults, including mothers?

Page 3: Consider changing “exposition” to “exposure”

Page 3: “…whether the child was under Cotrim” change to “whether the child received cotrimoxazole prophylaxis”.

Page 5: The authors emphasize the high rate of HIV positivity (16.09%) in children identified in inpatient settings, but the total number, 262, was small. The second largest source of children with HIV infection was the outpatient service, where the 11.12% positivity rate in 3198 children resulted in a far higher number of (newly?) identified children than all other sources besides PMTCT combined. It would be helpful to understand how logistically difficult it would be to introduce childhood HIV testing in all outpatient clinics in Cameroon.

Page 5: Welfare is misspelled in Tables 2, 3 and 4

Page 7: “consultation and hospitalization ward” – use terminology used throughout the manuscript – such as, “inpatient and outpatient services”

Page 7: “…as well as those whose mother did not even present at hospitals during pregnancy.” Is the only setting for testing women for HIV during pregnancy the hospital? Or is it that these women did not receive any care in formal medical settings (or whatever appropriate term is used for this care)?

Page 7: “…applied to all infants and children admitted to inpatient or outpatient wards…” Aren’t children seen in outpatient clinics – I am not sure what being admitted to an outpatient ward means, but seems more limited than the previous discussion of care settings. “…as well as to all adults.” This last phrase seems like a non sequitur. Improving the testing of women before they become pregnant or early in pregnancy so they can benefit from PMTCT services is an important corollary to the EID program, but this manuscript did not address issues relevant to testing all adults.

Comments on the Quality of English Language

Overall the manuscript was clear and easy to read. Only a few suggestions are offered. 

Reviewer 2 Report

Comments and Suggestions for Authors

In this study, Nguefeu and colleagues investigate at which point during a child's life they become diagnosed with HIV in Cameroon. Identifying when and under what care the diagnosis happens will inform strategies on how to most efficiently identify children, and their mothers, who are HIV-1 positive. From this study it is clear that a lot of children do not follow up with PMTCT care and that a significant number of children are identified as living with HIV when they enter the healthcare system via different services such as inpatient services, nutrition services, outpatient services, and the HIV care unit. This work is important - if you can't identify where the gaps in the health care system are to detect children who are living with HIV you cannot improve detection, provision of ART. This, therefore impact the population as a whole since HIV is transmitted both horizontally and vertically.

I have a couple of minor comment in section 3.3 and Table 3. In the text it is written that "a total of 219 (83.91%) were HIV-infected, while 42 (16.09%) were HIV-uninfected." The numbers in this sentence are swapped - the table is correct, 16.09% are infected rather than uninfected. 

In the discussion, it is written "The reach the first 95 in paediatrics" - what is meant by this? the first 95 what?

Comments on the Quality of English Language

Table 2 and table 3: it is infant welfare rather than welfare.

Table 4: title: HIV status according to the entry point after adjustment with other covariates.
